# Experimental Investigation of Acoustic Features to Optimize Intelligibility in Cochlear Implants

**DOI:** 10.3390/s23177553

**Published:** 2023-08-31

**Authors:** Fergal Henry, Ashkan Parsi, Martin Glavin, Edward Jones

**Affiliations:** 1Department of Computing and Electronic Engineering, Atlantic Technological University Sligo, Ash Lane, F91 YW50 Sligo, Ireland; 2Electrical and Electronic Engineering, University of Galway, University Road, H91 TK33 Galway, Ireland; ashk.parsi@gmail.com (A.P.); martin.glavin@universityofgalway.ie (M.G.); edward.jones@universityofgalway.ie (E.J.)

**Keywords:** acoustic features, cochlear implant (CI), machine learning (ML), neural network (NN), noise reduction (NR), speech enhancement (SE), speech intelligibility (SI)

## Abstract

Although cochlear implants work well for people with hearing impairment in quiet conditions, it is well-known that they are not as effective in noisy environments. Noise reduction algorithms based on machine learning allied with appropriate speech features can be used to address this problem. The purpose of this study is to investigate the importance of acoustic features in such algorithms. Acoustic features are extracted from speech and noise mixtures and used in conjunction with the ideal binary mask to train a deep neural network to estimate masks for speech synthesis to produce enhanced speech. The intelligibility of this speech is objectively measured using metrics such as Short-time Objective Intelligibility (STOI), Hit Rate minus False Alarm Rate (HIT-FA) and Normalized Covariance Measure (NCM) for both simulated normal-hearing and hearing-impaired scenarios. A wide range of existing features is experimentally evaluated, including features that have not been traditionally applied in this application. The results demonstrate that frequency domain features perform best. In particular, Gammatone features performed best for normal hearing over a range of signal-to-noise ratios and noise types (STOI = 0.7826). Mel spectrogram features exhibited the best overall performance for hearing impairment (NCM = 0.7314). There is a stronger correlation between STOI and NCM than HIT-FA and NCM, suggesting that the former is a better predictor of intelligibility for hearing-impaired listeners. The results of this study may be useful in the design of adaptive intelligibility enhancement systems for cochlear implants based on both the noise level and the nature of the noise (stationary or non-stationary).

## 1. Introduction

Audio is a term used to describe speech, music and, indeed, any sounds in the surrounding environment. There are various application areas associated with speech, such as speech recognition, speaker recognition, gender recognition, speech enhancement, speech-language pathology, voice activity detection, emotion recognition and blind source separation [1]. The application area of primary interest in this paper is speech enhancement, which is also known as noise reduction. Noise reduction can also be regarded as increasing speech intelligibility [2,3]. There has been extensive recent interest in the emerging area of machine learning-based speech enhancement techniques [4,5,6,7]. A particularly important application of speech intelligibility is its use in signal processors for cochlear implants (CIs) used by hearing-impaired listeners [8,9,10]. A key component of machine learning algorithms in such devices is the feature extraction block, whereby acoustic features are extracted from speech signals and are generally used as input with the intention of generating a higher quality or more intelligible version of the input speech. This paper presents a detailed analysis of a range of acoustic features for the purpose of improving speech intelligibility in the context of cochlear implant applications. The acoustic features considered in this study are organized into categories to facilitate their analysis. A wide range of existing features is considered, including features that have not traditionally been applied for enhancing intelligibility in cochlear implants. This study has looked at the limitations of previous work, built on it and extended it for this application. Furthermore, it is not the machine learning algorithm that is being evaluated here but rather how discriminative the features are in performing the task of separating speech from a noisy mixture.

This paper is structured as follows. Section 1 provides an introductory statement with an overview of what the paper is about. Section 2 gives more detailed background on previous work in speech enhancement in related areas and noise reduction in cochlear implants. The acoustic features under investigation, the related areas in which they have been previously applied and their associated parameters are presented in Section 3. The experimental setup, the speech corpus, the noise sources and speech intelligibility metrics are explained in Section 4. The speech intelligibility results are extensively documented and discussed in Section 5. Finally, the paper concludes with some closing remarks in Section 6. When comparing work done in this study to previous research, every effort has been made to highlight the similarities and, more importantly, differences.

The key contributions of this paper are as follows:It considers appropriate feature sets that have been successfully used in other audio application areas and carries out an experimental evaluation of how effective these features are at enhancing speech in the presence of noise, specifically for cochlear implant applications. This includes feature sets that have not been traditionally used in this particular application in the past.This study highlights which category of features is best in general for speech enhancement and, more specifically, which features are most appropriate to address stationary and non-stationary noises. The features are benchmarked in terms of speech intelligibility and processing time against a group of features, which have been previously presented in the literature [11].It discusses the possibility of a noise-adaptive speech enhancement system for a cochlear implant system, which is based on selecting the best feature type for each noise condition.This paper includes an experimental analysis of the amount of correlation between intelligibility metrics used for normal hearing and hearing-impaired situations.

## 2. Background

Moser [12] describes three different methods of hearing. The first of these is used by people with so-called “normal hearing”. Here, sound waves enter the ear canal, causing the eardrum/bones in the middle ear to vibrate. These vibrations cause the movement of sensory hair cells in the cochlea. As a result, neural signals travel along the cochlear nerve to the brain. The cochlea can be modeled as a spectrum analyzer as hair cells at different locations respond to different pitches of sounds. The other two methods of hearing are used by people with hearing impairment. One of these is referred to as “electrical hearing” and is used in the modern-day cochlear implant shown in Figure 1. Here, the microphone/s, speech processor and transmitter/encoder are placed external to the skin and usually behind the ear. The microphone acquires audio signals from the environment, the processor analyses these and the processed audio is encoded and transmitted wirelessly. The implanted receiver/decoder and the electrode array that is interfaced to the cochlea are internal to the skin. The processed audio is received and decoded before the electrode array delivers a stimulation pulse pattern to the cochlea. Unfortunately, the array can realistically only have a limited number of electrodes. Therefore, the pulses can stimulate adjacent nerve cells, resulting in a muddy sound that lacks clarity and definition. The final method is referred to as “optical hearing”. Here, the external hardware of a cochlear implant remains the same as previously described. However, the processor splits the sound into finer resolution frequency bands and hence can transmit a more precise stimulation pattern. As this is optical, a light source of choice is used, which spirals along the cochlea. The benefit of such an approach is an increased number of cochlear stimulation sites. This is because the light used in an optical cochlear implant can be more readily confined in space than the current used in an electrical cochlear implant.

The electrical cochlear implant performs reasonably well in quiet conditions without background noise, i.e., it allows a recipient to understand what is being said by a target speaker. However, the intelligibility degrades with increasing background noise. An approach to reduce this noise is to integrate a machine learning (ML) algorithm with pre-existing audio signal processing techniques. The effectiveness of any such algorithm depends on the careful selection of appropriate features, which are used to train/test the network used. Hence, feature extraction is a crucial part of any machine learning process introduced into a cochlear implant system.

The literature on audio signal processing is well-summarized by Sharma et al. in [1], with a special focus on feature extraction techniques. They categorize features into the following domains: temporal, frequency, time-frequency, cepstral and wavelet. They also highlight that the following features are suitable for speech recognition/enhancement applications: Linear Predictive Coding Coefficients (LPC), Linear Predictive Coding Cepstral Coefficients (LPCC), Mel Frequency Cepstral Coefficients (MFCC), fundamental frequency, skewness, kurtosis, entropy, Perceptual Linear Prediction Coefficients (PLP), Relative-Spectral Transformed PLP Coefficients (RASTA-PLP), Gammatone Cepstral Coefficients (GTCC), wavelet and sparse features. Alias et al. [13] also reviewed feature extraction as used in audio signal processing and initially organized the approaches as being either physical or perceptual. Then, they subdivided these categories further into time, frequency, wavelet, image-based, cepstral and other domains.

Some approaches in audio signal processing decompose signals prior to feature extraction. The Discrete Wavelet Transform (DWT) and The Discrete Wavelet Packet Transform (DWPT) have been widely employed for this task. Van Fleet provides a rigorous mathematical explanation of these transforms in [14].

Tzanetakis et al. [15] used the mean, standard deviation and ratios of coefficients in each sub-band of the DWT, combining them with MFCC and Short Time Fourier Transform (STFT) features to perform three separate classification tests. The first was speech and music, the second was male and female, and the third was choir, orchestra, piano and string quartet. Separately, a wavelet-based feature vector was proposed by Kumari et al. [16] for the purpose of audio signal classification. This vector was composed of mean and standard deviation values of sub-band power, pitch, brightness, bandwidth and frequency cepstral coefficients. Ali et al. [17] applied a three-level DWT and used the normalized energy in each sub-band to generate features to perform automatic speech recognition of Urdu. Thiruvengatanadhan [18] used the DWT to classify speech and music and found that the Daubechies8 (db8) wavelet performed better than either Haar or Symlets2.

Gowdy and Tufekci [19,20] also used the DWT to perform speech recognition tasks. They compared different feature extraction techniques, namely MFCC, sub-band features, multiresolution features and Mel-frequency discrete wavelet coefficients. Subsequently, they added the log-normal approximation approach to the Parallel Model Combination (PMC) technique in order to improve performance in noise [21,22]. Abdalla et al. [23] performed a three-level DWT before extracting MFCCs from wavelet channels. Then, they used Hidden Markov Models (HMMs) for speaker identification. Later, they replaced the HMMs with a neural network for speech recognition and found that the wavelet-based MFCCs performed better at speech recognition for isolated words than using MFCCs alone [24].

Some interesting issues arise when replacing the DWT with the DWPT. Zou et al. [25] applied it to extract the vibroacoustic signature of diesel engines and found that they had to use a wavelet packet shifting theorem when sorting the packets in ascending order of frequency. Ariananda et al. [26] evaluated it as a method for spectrum estimation and highlighted the fact that when ordering the packets in terms of frequency, one must observe a binary Gray code sequence. Kobayashi et al. also addressed the shift-invariance and frequency order of the DWPT in [27].

Anusuya and Katti [28] extracted LPC, MFCC and RASTA-PLP features with and without wavelet packet decomposition. They achieved the best performance in noisy conditions for Kannada speech recognition when they used RASTA-PLP with a five-level DWPT and db8 wavelet. Nehe and Holambe [29] used white noise as the background and extracted dyadic wavelet decomposed LPCs from a three-level DWT and uniform wavelet decomposed LPCs from a two-level DWPT for the purpose of isolated word recognition. They found both methods to be efficient and achieved benefits in memory requirement and computation time. Magre et al. [30] found that extracting LPC features following a three-level decomposition using the DWT provided better speech recognition accuracy than directly extracting LPC and MFCC features. Turner and Joseph [31] investigated three different methods for computing MFCCs to perform speaker identification. They used a Gaussian Mixture Model (GMM) to model the speech along with the Discrete Fourier Transform (DFT), DWT and DWPT for extracting acoustic features. Feng and Wang [32] performed speech emotion recognition, which was based on a Long Short-Term Memory (LSTM) network and Mel scale wavelet packet decomposition. They extracted features such as sub-band energy, MFCC, formant, pitch and short-term energy.

A typical acoustic scene contains a mixture of the desired target speech and various unwanted background sounds. Speech enhancement involves the extraction of the clean target from this mixture. However, visual information is also available, which can be exploited by the target speaker. Two of the more obvious visual cues are lip movements and facial expressions. Adding this additional information could further improve speech enhancement systems. Michelsanti et al. [33] systematically researched this topic and have identified the following as the main areas of interest: acoustic features, visual features, deep learning, fusion, training targets and objective functions.

As outlined earlier, CI users have difficulty understanding speech in the presence of background noise, especially of the non-stationary type, e.g., babble. Because of the background noise, they are less able to discern temporal fluctuations. The spectral resolution of their devices is limited by the number of electrodes to compound matters, as already outlined [34]. CI devices have previously employed single-channel speech enhancement algorithms such as spectral subtraction and Wiener filtering. They have somewhat improved speech intelligibility in the presence of stationary noise, such as speech-shaped noise. However, they have proved less effective at improving intelligibility in non-stationary noise. Consequently, researchers such as Goehring et al. [35] promoted the use of data-driven algorithms employing machine learning techniques, such as neural networks (NNs) and Gaussian Mixture Models (GMMs), to enhance speech more successfully when mixed with non-stationary noises. Another prime example of a neural network speech enhancement (NNSE) system for cochlear implants was implemented by Goehring et al. [36], where they used a feature vector composed of Gammatone Frequency Energy, Gammatone Frequency Cepstral Coefficients and Gammatone Frequency Perceptual Linear Prediction Cepstral Coefficients.

While the main focus of this paper is the choice of acoustic features used in speech enhancement for cochlear implants, the choice of machine learning architecture and, indeed, the masking/training target is also important, for example, as outlined in [37,38]. A range of architectures has been used, as outlined in the studies by Wang et al. [39] and Nossier and colleagues [40]. The results of the latter study demonstrate that there is no ideal training target that satisfies all quality/intelligibility metrics. As usual in engineering, there is a trade-off between the method of speech enhancement and the intelligibility of the output speech. Seleem et al. [41] trained deep neural networks (DNNs) and recurrent neural networks (RNNs) to learn spectral masking from magnitude spectrograms of noisy speech. Subsequently, Abdullah et al. [42] proposed a Quantized Correlation Mask to improve the efficiency of a DNN-based speech enhancement system, which previously employed the Ideal Ratio Mask. Separately, Roy et al. [43] suggested an improved method of enhancing speech called the DeepLPC technique. Their method estimated clean speech along with noise linear prediction coefficient (LPC) parameters for the augmented Kalman filter (AFK).

It is useful to simulate speech processing in a cochlear implant to avoid the need for subjective listening tests with CI users in this study. If one can do so successfully, one can mimic the sounds heard by CI users. A vocoder essentially allows us to do so by acting as a voice processing system, which is capable of analyzing and resynthesizing speech signals. Previously, it has been used in applications such as audio data compression [44], voice encryption [45], voice modification [46] and even as an electronic musical instrument [47]. Researchers have adapted it as a tool to simulate the operation of a cochlear implant. Such simulations have been used to predict speech quality and, indeed, intelligibility for CI users. One can present vocoded sounds to normal-hearing people for the purpose of subjective listening tests. Using a vocoder simulation, one can readily vary parameters such as noise type, speech masker type, signal-to-noise ratio (SNR), number of electrodes/channels, etc. This allows us to objectively estimate the effect of such changes on speech quality and, more importantly, intelligibility. Although such simulations cannot absolutely predict intelligibility performance, they can suggest a trend as these parameters are adjusted. One of two possible types of vocoder is illustrated in Figure 2. In this noise vocoder, white noise acts as a carrier, and the amplitude-modulated noises from all channels are summed at the output. Alternatively, a tone vocoder can be used where white noise is replaced by sinusoidal carriers. The center frequencies of carriers correspond with those of the band-pass filters. As can be seen, baseband signals in each channel are amplitude-modulated with the carrier and all the modulated signals are summed, giving the vocoded speech signal at the output. Either type of vocoder can be used to estimate speech quality or intelligibility. The vocoder has been extensively used in many recent studies related to cochlear implants, such as [35,36,48,49,50]. This paper uses the noise vocoder to simulate speech processing in a cochlear implant.

In tandem with developments in signal processing in cochlear implants, there have also been developments in other components of technology to assist the hearing impaired. Smart healthcare utilizes technology (e.g., wearable devices, Internet of Things, mobile communications) to acquire and manage healthcare information. It can also connect people working in healthcare, materials and institutions and can actively manage and respond in an intelligent fashion to the requirements of medical ecosystems [51]. In addition, machine learning techniques have been employed in cochlear implants and various other systems [52,53,54,55] that can be embedded in portable devices in the smart healthcare domain. Various approaches to machine learning in healthcare were reviewed by Rayan and colleagues [56]. They examined datasets, preprocessing, feature extraction and machine learning techniques used in applications such as Glaucoma diagnosis, prediction of Alzheimer’s disease, bacterial sepsis diagnosis, prediction of intensive care unit (ICU) readmissions and cataract detection. More specifically, in the context of our work, Crowson et al. [9] conducted a structured review of machine learning in cochlear implants. Their survey showed that the machine learning algorithms used for the various applications encompassed neural networks, support vector machines, Gaussian Mixture Models and linear regression. A major challenge in the field is to replace centralized cloud servers with alternative microcontroller unit (MCU) inference devices, which are closer to the platform of the sensors in healthcare wearables. This, in turn, requires consideration of computational requirements. For this purpose, Diab et al. [57] conducted research into Tiny Machine Learning (TinyML), which allows machine learning algorithms to be implemented on embedded devices with serious resource constraints. They concluded that this approach results in better latency, power and privacy than cloud computing. A recent illustration of machine learning in cochlear implants is where Alohali et al. [58] were able to predict the impedance of the electrode arrays in each channel of cochlear implants with a MED-EL FLEX 28 electrode array at different time points after the date of surgery. They used different machine learning algorithms such as linear regression, Bayesian linear regression, decision forest regression, boosted decision tree regression and neural networks.

## 3. Features

This section discusses the feature sets used in this study. For later sections, each feature is identified by an abbreviation which is introduced here in brackets after the feature name (see also Table 1). Given the range of features examined, the dimension of the feature vector (including delta features) ranged in size from 22 to 512. All speech signals used in the experiments were framed into 20 ms segments. The frames were subsequently overlapped by 50% before the features were extracted. In order to provide temporal context, features have additionally been spliced into five-frame windows. Useful information on feature transition has been gained by concatenating instantaneous features with associated delta features. The features evaluated in this study have been categorized according to the approach used by Sharma and colleagues [1]. Numerous features were initially tested to determine STOI scores using factory noise from the Signal Processing Information Base (SPIB) [59] as the interference at −5 dB SNR. When optimizing a particular feature parameter (e.g., number of Linear Predictive Coding Cepstral Coefficients), the STOI intelligibility score was computed for various parameter values. Then, the optimal value of the parameter was chosen based on the peak STOI score. The final features were shortlisted for this study based on their effectiveness in related audio applications and their preliminary STOI scores.

### 3.1. Frequency Domain Features

The first category considered here is frequency domain features. When performing frequency analysis of a signal, the time to frequency domain conversion is typically performed using either the Discrete Fourier Transform (DFT) or auto-regression.

Three commonly used scales in the field of psychoacoustics are Bark, Equivalent Rectangular Bandwidth (ERB) and Mel [60]. The Bark scale is a psychoacoustic measurement of frequency and reflects the fact that there are linear/logarithmic responses to lower/higher frequencies, respectively. The basilar membrane of the inner ear essentially behaves like a spectrum analyzer, where each point on it acts as a band-pass filter. The bandwidth of each filter can be regarded as being equal to one critical bandwidth. Here, a critical bandwidth is referred to as a Bark. The Bark scale covers approximately 24 critical bands for normal hearing. The ERB scale also provides approximate bandwidths for the band-pass filters in human hearing. The Mel scale is a perceptual scale of pitch, which imitates the nonlinear perception of sound in the human ear, i.e., it is more/less discriminative at lower/higher frequencies, respectively. Various spectra using these scales have been generated to extract frequency domain features. The audio feature extractor tool in Matlab R2021b was utilized to extract 32 Bark Spectrum (BS), 34 ERB Spectrum (ES) and 32 Mel Spectrum (MS2) features.

Various other spectra were generated as follows. A 64-channel Gammatone filter was used to extract Gammatone Filterbank Power Spectra (GF) features. Separately, 35 Octave Spectrum (OS) features were extracted. In this case, the Octave Spectrum was computed by averaging the power over octave bands as specified by the ANSI S1.11 standard [68]. The number of bands per octave was six, the frequency ranged from 150 Hz to 8 kHz and ‘A’ weighting was used. The final spectrum had 32 features, and the Short Time Fourier Transform (STFT) was used to extract them. The length of the FFT was 512, and a one-sided STFT was computed. Frequency ranged from 50 Hz to 8 kHz and was divided into 32 overlapping bands according to the Mel scale. No triangular filtering was applied in this case. Instead, the energy within each band was calculated, and the logarithm was taken for the energies before applying the Discrete Cosine Transform (DCT).

The cochleagram is similar to the spectrogram, except a gammatone filter is used instead of linearly spaced center frequencies [74]. In this study, four 64-channel cochleagrams at different spectro-temporal resolutions were combined into what is referred to as a Multiresolution Cochleagram (MRCG). The 256 MRCG features are considered to capture local as well as contextual information according to the work of Chen et al. in [67]. Separately, a Mel Spectrogram (MS1) was generated with 24 features using an algorithm similar to that described in [65].

Spectral shape is another type of feature that can be calculated in the frequency domain. This feature type has been reviewed by Al-Shoshan [71] for the purpose of speech and music classification. For this study, eleven descriptors were used to represent spectral shape (SS): spectral centroid, spectral roll-off frequency, spectral spread, spectral skewness, spectral kurtosis, spectral slope, spectral decrease, spectral flatness, spectral crest, spectral entropy and spectral flux [1]. The spectral centroid is the center of mass of a spectrum. The spectral roll-off point is the frequency below which ninety-five percent of spectral energy is contained. The bandwidth of a signal and its spectral spread are closely related. The symmetry of a spectrum around its arithmetic mean is referred to as its spectral skewness. How flat a spectrum is around its mean is known as its spectral kurtosis. The spectral slope of the amplitude of a signal can be calculated by performing linear regression, while averaging the spectral slope at low frequencies determines the spectral decrease. Determining how uniform the frequency distribution of a power spectrum is a measure of its flatness. The spectral crest factor is a measure of the peaked nature of the power spectrum of a signal. Spectral entropy is a measure of how uniformly flat a spectrum is. Spectral flux may be calculated using Equation (1). Here, *s(k)* is the spectral value at bin *k*, *b*_1_ and *b*_2_ are the band edges, *t* is the center of the current frame, *t* − 1 is the center of the previous frame and *p* is the norm type.
(1)$$specFlux\left(t\right)={\left(\sum_{k=b_{1}}^{b_{2}}{\left|s_{t}\left(k\right)-s_{t-1}\left(k\right)\right|}^{p}\right)}^{\frac{1}{p}}$$

### 3.2. Cepstral Domain Features

The second category considered here includes a variety of cepstral domain features. A cepstrum may be calculated by performing the Inverse Fourier transform (IFT) of the logarithm of the spectrum of a signal. This can produce four types of cepstra, namely complex, power, phase and real. As speech processing is being performed in this study, the power cepstrum was deemed to be the most appropriate.

The first type of cepstral coefficients considered here were Constant Q Cepstral Coefficients (CQCC). In this study, one zeroth and 19 static CQCCs were extracted as described by Todisco and colleagues [61]. The minimum frequency was calculated as 15 Hz using Equation (2), assuming a maximum frequency of 8 kHz. The Nyquist frequency is *f_NYQ_*, nine octaves were used and the number of bins per octave was 96.
(2)fmin=fmax29=fNYQ229

Another type of cepstral feature is Gammatone Cepstral Coefficients (GTCCs). They were used by Yin et al. [63] in the area of speech recognition and hence were investigated as a possible candidate for this study. Gammatone filterbanks were employed here to generate a cochleagram to represent the various sounds in the time-frequency domain. Consequently, 31 GTCCs were extracted. In this study, the best results were achieved when using 30 channels, appending the log energy, selecting a frequency range from 50 Hz to 8 kHz and using cubic-root rectification.

Linear Predictive Coding (LPC) coefficients are often categorized as frequency domain features. However, they can be adapted to generate Linear Predictive Cepstral Coefficients (LPCC). LPCCs have been previously used by Chen et al. for noise removal [64]. After experimenting with different orders for this study, optimum intelligibility was achieved when extracting 20 LPCCs.

An obvious choice of feature type to extract from the cepstra of the speech signals under test is Mel Frequency Cepstral Coefficients (MFCCs) [1]. They can be calculated by taking the Discrete Cosine Transform of the log power spectrum of the signal on a nonlinear Mel scale. MFCCs were previously used for classifying environmental noise for CI users in [66]. A 31-dimensional MFCC feature vector was used in this study.

Greenwood functions are a generalized form of MFCCs. The idea of the Greenwood equation is to map the cochlear-frequency position function for all species of mammals. In this study, 64 channels were used to extract 31 Greenwood Function Cepstral Coefficients (GFCCs). The method used is well-described in the work by Clemins and colleagues [62]. The minimum and maximum frequencies used were 62.5 Hz and 8 kHz, respectively. The perceived frequency is calculated using Equations (3)–(5) using a value of 0.88 for *k*.
(3)$$A=\frac{f_{min}}{1-k}$$
(4)$$a=\log_{10}\left(\frac{f_{max}}{A}+k\right)$$
(5)fp=1alog10⁡fA+k

Hermansky was one of the pioneers of Perceptual Linear Prediction (PLP) analysis of speech. His equal-loudness-curve formula in Equation (6) has been used here to approximate the unequal sensitivity of human hearing across frequencies [69]. The PLP coefficients were obtained from the LPCs by performing perceptual processing and then applying an auto-regressive model. Consequently, linear coefficients were transformed into more useful cepstral coefficients. The best results were obtained in this study when 12 Perceptual Linear Prediction Coefficients (PLP) features were extracted.
(6)$$eql={\left(\frac{f^{2}}{f^{2}+1.6e5}\right)}^{2}\left(\frac{f^{2}+1.44e6}{f^{2}+9.61e6}\right)$$

### 3.3. Wavelet Domain Features

The third and final category considered in this study was wavelet domain features. It is possible to transform audio from its time to time-frequency representation using the Discrete Wavelet Transform (DWT). The DWT was considered for this study as an alternative to the STFT to extract information from more challenging non-stationary signals. An advantage of this transform is that it gives high time and low-frequency resolution for higher frequencies and vice versa. The DWT generates both approximation and detailed coefficients, both of which can be used as wavelet features. One can extract the features using either the basic DWT or the Discrete Wavelet Packet Transform (DWPT) [1]. When using the DWT, only the approximation coefficients at each level were further decomposed. On the other hand, both approximation and detailed components were decomposed when employing the DWPT. Features previously described, such as MFCC and PLP, can then be subsequently extracted from the wavelet packet decomposition.

Dai et al. [70] performed a four-level Sparse Discrete Wavelet Decomposition (SDWD) with an eight-tap Daubechies filter to extract 10 features for the purpose of speech recognition. Their work, which used the DWPT, was adapted further in this study to use five levels of decomposition with the Fejér-Korovkin (fk22) wavelet to extract 14 SDWD features. Unlike Dai et al., we do not use a voice activity detection algorithm in speech preprocessing. Figure 3 illustrates the relationship between the fk22 scaling function and its wavelet. It is useful for application in the DWPT due to its orthogonal nature and the fact that its filter construction minimizes the difference between a scaling filter and the ideal low-pass sinc function.

Palo and Kumar [72] extracted 16 Wavelet Mel Frequency Cepstral Coefficients (WMFCC) when investigating the recognition of emotions. This study built on that work using the DWT with the fk22 wavelet and, in this case, three levels of decomposition. Twenty WMFCCs were extracted in this study. Best intelligibility was achieved by extracting four coefficients from the approximation coefficients at level 3 (representing frequencies ranging from 0–1 kHz), six coefficients from the detail coefficients at level 3 (representing frequencies ranging from 1–2 kHz), six coefficients from the detail coefficients at level 2 (representing frequencies ranging from 2–4 kHz) and four coefficients from the detail coefficients at level 1 (representing frequencies ranging from 4–8 kHz). It should be noted that Palo and Kumar used utterances from a different database, i.e., Surrey Audio-Visual Expressed Emotion (SAVEE), and unlike them, we did not perform any feature reduction or vector quantization.

Pursuing the work of Gaafar et al. [73], 20 Wavelet Perceptual Linear Prediction Coefficients (WPLP) were extracted. In this work, the best order of the PLP model for each level of decomposition was found to be four. Again, the DWT was used in conjunction with the optimum fk22 wavelet and three levels of decomposition. It should be noted that Gaafar et al. used only one hidden layer, whereas we used three. They used the db7 wavelet as opposed to our fk22. Additionally, they extracted RASTA-PLP features, whereas we extracted PLP features as these achieved better intelligibility scores in our application.

## 4. Methodology

### 4.1. Training and Test Data

The dataset used for this work contains 600 utterances of speech mixed with noise for training and 120 mixtures for testing. The mixtures were obtained by mixing clean utterances with babble and speech-shaped noise (SSN) separately. Mixing was done at –5 dB, 0 dB and 5 dB to test low, medium and high signal-to-noise ratios (SNRs), respectively. All 720 sentences were taken from the IEEE-Harvard corpus [75] and were produced by a male talker. All signals were sampled at 16 kHz. This corpus was chosen as the sentences are phonetically balanced and are widely used in the research of speech and acoustics, such as in [2]. Babble and SSN were chosen as representative examples of non-stationary and stationary noise, respectively. Babble noise was sourced from SPIB [59]. SSN was generated by filtering white noise from SPIB using the coefficients of the long-term speech envelope (LTSE) of the speech corpus. The first 100 utterances from the corpus were concatenated to implement this approach. Both noises were approximately four minutes in duration. Training and test sets were generated as per the work conducted in [11]. Training sets were generated by mixing random cuts from the first two minutes of each noise type with clean training utterances at SNRs of −5 dB and 0 dB. Test sets were generated by mixing random cuts from the last two minutes of each noise with the clean test utterances at SNRs of −5 dB, 0 dB and 5 dB. The noises were divided into two equal-sized groups, ensuring that new noise segments were presented during the test phase.

### 4.2. Speech Enhancement System

The machine learning architecture used is based on that used by Wang and colleagues [11]. The choice of this architecture is based on the fact that it has been extensively studied in previous literature; as noted earlier, the objective of this paper is to evaluate the features, not the machine learning architecture; therefore, the use of a single machine learning architecture that can be considered a well-established benchmark system in this paper is appropriate. Open-source software for its implementation can be found at http://web.cse.ohio-state.edu/pnl/DNN_toolbox/ (accessed on 9 October 2020).

Multilayer perceptrons are used as the DNNs, which are trained across 64 frequency bands with the mean squared error (MSE) used as the cost function. The neural network used for enhancement contains three hidden layers, each containing 1024 nodes. Sigmoid activation functions were used for both hidden and output layers. After feature extraction, in order to provide smoothing of temporal trajectories of features, an auto-regressive moving average filter of order 2 was used. The overall test architecture is depicted in Figure 4. The inputs used were the clean utterances and the babble/SSN noise. These were mixed at the appropriate SNRs to generate the noisy signals. These mixtures were then loaded into a gammatone filter with 64 channels, as per Wang and colleagues [76]. The center frequencies of this filter were equally spaced between 50 Hz and 8 kHz on the ERB scale. The 64-channel outputs were framed as previously described, resulting in cochleagrams of the mixtures. The network was trained with the backpropagation algorithm, and no unsupervised pretraining was used. Dropout can be used to address overfitting in neural networks. However, for this study, which is a comparison of acoustic features, no dropout was used. One potential downside of dropout is that it can sometimes miss important data trends where applications use time-series data, and the use of dropout with single samples or small groups could cause important training data to be missed. The number of training epochs was set to 20. Optimization was achieved with adaptive gradient descent, and a momentum term was also used. The momentum term was set to 0.5 for epochs 1 to 5 and was assigned a value of 0.9 thereafter. As is common when working on a regression model, the mean squared error (MSE) was utilized as the cost function. Sigmoid activation functions were used in the output layer and hidden layers. Ideal masks and estimated masks were generated accordingly. The estimated masks were used in conjunction with the noisy signals to synthesize signals, which are “enhanced” estimations of clean speech. This is regarded as normal speech, and hence, STOI is used to calculate its intelligibility. The estimated clean speech must be further vocoded to simulate speech as heard by a cochlear implant user. Only then can NCM scores be calculated to estimate the intelligibility of enhanced speech for hearing-impaired listeners.

### 4.3. Training Target

The training target selected for this study is the Ideal Binary Mask (IBM) [77]. It is mathematically defined as follows:(7)$$IBM\left(t,f\right)=\left\{\begin{array}{c} 1,\;if\;SNR\left(t,f\right)>LC\\ 0,\;otherwise \end{array}\right.$$

In Equation (7), *SNR(t,f)* is the local SNR within a time-frequency (T-F) unit at time *t* and frequency *f*. A local criterion (*LC*) of 5 dB less than the mixture SNR has been chosen for this study as it was found by [11] to still preserve sufficient speech information. For this work, the IBM was generated using a 64-channel Gammatone filterbank rather than an alternative approach, which uses the STFT. Other masks, such as the Ideal Ratio Mask (IRM), may increase intelligibility, but in the context of this study on features, the IBM has been chosen as it is well-established in benchmarking speech enhancement algorithms.

### 4.4. Speech Segregation

Supervised speech separation is performed in this study. Here, neural networks were trained using acoustic features from mixtures as well as the matching desired outputs. Speech segregation is also known as speech separation. During testing, different mixtures were segregated using an estimated mask. The separated speech can be resynthesized using the estimated output magnitude and the phase of the original mixture. A typical separation/segregation algorithm was suggested by Healy et al. [78] and is depicted in Figure 5. The training mentioned above can be performed with a binary target. However, a soft mask is used for testing where the posterior probabilities from the NN represent the probability of the dominance of speech.

### 4.5. Cochlear Implant Simulation

A noise-vocoder with eight channels (Figure 2) was used as the CI simulation system. Initially, normal speech signals were enhanced using the NR stage in this study. The segregated speech was then vocoded by the noise vocoder. The vocoder has five stages, namely pre-emphasis, band-pass filtering, envelope detection, modulation and band-limiting stage. The pre-emphasis filter specifications were as follows: cut-off frequency 2 kHz and roll-off 3 dB/octave. After pre-emphasis, the BPFs filter the resultant signals into eight frequency bands (channels) with center frequencies of 366, 526, 757, 1089, 1566, 2252, 3241 and 4662 Hz. Full-wave rectifiers are cascaded with low-pass filters with a cut-off frequency of 160 Hz to extract the temporal envelopes in each band. These envelopes are then amplitude-modulated with white noise. Further filtering is carried out with the same set of BPFs as in the previous stage. The eight modulated signals are added together. Finally, the level of this signal is adjusted to match that of the rms value of the input signal to the vocoder, as recommended in [79]. Open-source software for the vocoder can be found at https://github.com/vmontazeri/cochlear-implant-simulation (accessed on 1 November 2020).

### 4.6. Intelligibility Metrics

Separate metrics were considered here for both normal and vocoded speech. Two metrics were used here for estimating the intelligibility of normal speech, namely STOI and HIT-FA. As IBM was used here as the training target, the HIT-FA rate is an appropriate evaluation metric according to the work of [39]. HIT is the percentage of speech-dominant T-F units in the IBM that are correctly classified, whereas FA (false-alarm) is the percentage of noise-dominant T-F units that are wrongly classified. This metric is said to correlate well with speech intelligibility (SI). The other normal speech metric used was Short-time Objective Intelligibility (STOI). It is a measure of the correlation between short-time temporal envelopes of a clean utterance and its enhanced form. Typically, its value varies between 0 and 1, which can be interpreted as between 0% and 100% intelligibility. Although it may over-predict intelligibility, it nonetheless appears to correlate well with human intelligibility [39]. Finally, the Normalized Covariance Measure (NCM) is used to estimate the intelligibility of speech, which has been processed and then vocoded. Here, the reference signal is the clean vocoded speech. NCM is a speech transmission index-related measurement where the covariance of envelopes between the clean vocoded and processed vocoded signals is estimated. Because there are similarities between the method of computation of NCM and cochlear implant processing strategies, it has been found that the NCM measure correlates well with the SI of vocoded speech [49,80]. The calculation of the NCM is outlined in Equations (8)–(11). Here, *r_i_* is the normalized covariance in the *i*th frequency band, *x_i_(t)* is the downsampled envelope of the clean speech, *y_i_(t)* is the downsampled envelope of the processed speech, *µ_i_* is the mean value of *x_i_(t)* and *v_i_* is the mean value of *y_i_(t)*. $\bar{SNR}$ is the apparent SNR in each band. *TI_i_* is the transmission index in each band. *W_i_* are the band-importance weights, which are applied to bands 1 to *K*.
(8)$$r_{i}=\frac{\sum_{t}\left(x_{i}\left(t\right)-\mu_{i}\right).\left(y_{i}\left(t\right)-v_{i}\right)}{\sqrt{\sum_{t}{\left(x_{i}\left(t\right)-\mu_{i}\right)}^{2}}.\sqrt{\sum_{t}{\left(y_{i}\left(t\right)-v_{i}\right)}^{2}}}$$
(9)$$\bar{SNR}=10\;\log_{10}\left(\frac{r_{i}^{2}}{1-r_{i}^{2}}\right)$$
(10)$${TI}_{i}=\frac{\bar{SNR}+15}{30}$$
(11)$$NCM=\frac{\sum_{i=1}^{K}W_{i}{TI}_{I}}{\sum_{i=1}^{K}W_{i}}$$

## 5. Results and Discussion

When a standard normal Z-distribution is used and the respective values of 95%, 0.05 and 0.01 for the confidence levels, population standard deviation *σ* and margin of error *δ* are assumed, the minimum sample size *n* can be estimated according to (12) as being greater than or equal to 96 [81]:(12)$$n\geq {\left(\frac{1.96\sigma }{\delta }\right)}^{2}$$

In this study, this criterion is satisfied as the mean score and standard deviation for both STOI and NCM are calculated over 120 utterances in the test set. The higher the mean score and the lower the standard deviation, the better. Performance is evaluated for both “normal” speech (speech that has not been processed by a vocoder) using STOI and HIT-FA, as well as vocoded speech using NCM. This allows consideration of the best features for normal-hearing and hearing-impaired users. The scores for the best-performing feature sets in the tables are highlighted in bold, while those for the worst-performing feature sets are in italics. The results of this study are benchmarked against the feature set of Wang et al. under the same test conditions. The major difference is that Wang et al. [11] used a 246-dimensional feature vector composed of 15 AMS (amplitude modulation spectrogram) features, 13 RASTA-PLP features, 31 MFCC features, 64 GF features and corresponding delta features. They also used the TIMIT speech corpus and the Noisex dataset for the utterances and noises, respectively, while we used the IEEE Sentence corpus for consistency with similar work in [2].

The following sub-sections present the results for normal speech and vocoded speech. As noted above, features are identified by the acronyms listed in Table 1.

### 5.1. Results for Normal Hearing Case (STOI)

Table 2 presents the STOI scores for babble and SSN noises. The STOI scores for SSN are better than those for babble as it is more challenging for people with normal hearing to listen in the presence of non-stationary noise. BS is best at −5 dB, and GF is best at 0 and 5 dB for babble. LPCC features are consistently worst for babble. ES is best at −5 dB, and GF is best at 0 and 5 dB for SSN. SS is consistently worst for SSN. At −5 dB SNR, the standard deviation of the STOI scores when using the features for noise reduction is greater than the standard deviation of the STOI scores for the unprocessed mixtures. However, as the SNR increases, speech enhancement improves, and hence the roles are reversed.

It can be seen that the mean value of speech intelligibility increases, and its variation decreases as SNR increases. The BS feature mean STOI score for babble noise at −5 dB SNR is 0.6309. This is slightly better than the benchmark of 0.6235. Its variation of 0.0666 is slightly better than the benchmark variation of 0.0689. Compared with the benchmark, the feature vector is reduced in dimension from 246 to 64. The ES feature mean score for SSN at −5 dB SNR is 0.7188. This is slightly better than the benchmark of 0.7037. Its variation of 0.0562 is slightly better than the benchmark variation of 0.0576. Compared with the benchmark, the feature vector is reduced in dimension from 246 to 68.

It is worth considering whether there is any additional benefit in terms of intelligibility by performing wavelet decomposition and then carrying out further analysis on the resultant coefficients/packets. The score of 0.6168 for directly calculating MFCC is marginally better than using the wavelet approaches of SDWD (0.6156) and WMFCC (0.6150) for −5 dB babble. The corresponding score of 0.6214 for directly calculating PLP is marginally better than using the wavelet approach of WPLP (0.6188). The score of 0.7013 for directly calculating MFCC is marginally worse than using the wavelet approaches of SDWD (0.7059) and WMFCC (0.7032) for −5 dB SSN. The corresponding score of 0.7010 for directly calculating PLP is marginally worse than using the wavelet approach of WPLP (0.7097). Given these small differences, there does not seem to be any significant difference in STOI observed by using wavelet decomposition.

Figure 6 shows the improvement in mean STOI at −5 dB SNR relative to the unprocessed mixture (UN) for babble and SSN. This is only plotted at −5 dB SNR as this is the most challenging condition tested in this study for people with normal hearing. As can be seen, improvement is greater for SSN and varies between 0.1043 and 0.1339 across the range of features examined. Expressed as a percentage improvement in STOI, this corresponds to a range from 17.83% for SS to 22.89% for ES. The improvement for babble is less dramatic and ranges from 0.0378 to 0.071. Expressed as a percentage improvement in STOI, this corresponds to a range from 6.75% for LPCC to 12.68% for BS.

Figure 7 shows how STOI increases as SNR increases for both babble and SSN. The feature types plotted here were selected based on worst- and best-performing mean STOI scores. When the STOI scores are further averaged for each feature type over all 3 SNRs and both noise types, the GF features are best at 0.7826. These are closely followed by ES features at 0.7814, while LPCC features are the least effective at 0.7596.

### 5.2. Results for Normal Hearing Case (HIT-FA)

The HIT-FA scores for normal speech in the presence of babble and SSN are shown in Table 3 for SNR values of −5, 0 and 5 dB. The HIT-FA scores for SSN are better than those for babble for the same reason as outlined previously. Table 3 shows that the MRCG is best at −5 dB, and GF is best at 0 and 5 dB babble. SS features are consistently worst for babble. GF is best at −5 dB and 5 dB, while MRCG is best at 0 dB for SSN. GFCC is worst at −5 dB, while SS is worst at 0 and 5 dB for SSN. The MRCG feature HIT-FA score for babble noise at −5 dB SNR is 48.95%. This is slightly better than the benchmark of 48.24%. However, compared with the benchmark, the feature vector is increased in dimension from 246 to 512. The GF feature HIT-FA score for SSN at −5 dB SNR is 69.20%. This is slightly better than the benchmark of 67.70%. Compared with the benchmark, the feature vector is reduced in dimension from 246 to 128.

The training epoch that results in the best-performing model (selected based on performance on the validation set during training) is contained within the brackets in Table 3. As some of the best performances for HIT-FA were recorded at epoch 20, it may be beneficial to increase the number of training epochs from 20 to, say, 25 in future work. The HIT-FA scores here for babble noise at −5 dB SNR favorably compare with those of Wang and colleagues [39]. The test conditions were similar for both, and the features, which are common to both studies, rank in order of increasing HIT-FA as PLP, MFCC, GF and MRCG.

The score of 45.46% for directly calculating MFCC is marginally better than using the wavelet approaches of SDWD (42.28%) and WMFCC (43.76%) for −5 dB babble. In contrast, the corresponding score of 43.52% for directly calculating PLP is marginally worse than using the wavelet approach of WPLP (45.40%). The score of 63.35% for directly calculating MFCC is marginally better than using the wavelet approaches of SDWD (62.23%) and WMFCC (61.83%) for −5 dB SSN. The corresponding score of 64.24% for directly calculating PLP is marginally better than using the wavelet approach of WPLP (63.23%). The benefit or otherwise of using wavelet decomposition for HIT-FA seems to depend on which feature set is being utilized.

Figure 8 shows the HIT-FA scores (progressing from worst to best on the horizontal axis) for all features at −5 dB SNR for babble and SSN. The variation in HIT-FA improvement is between 36% to 49% for babble and between 58% and 69% for SSN. The MRCG and GF features are generally the strongest performers.

Figure 9 shows how HIT-FA increases as SNR increases for both babble and SSN. The feature types plotted here were selected based on the worst and best-performing HIT-FA scores. When the HIT-FA scores are further averaged for each feature type over all 3 SNRs and both noise types, the GF features are best at 68.92%. These are closely followed by MRCG features at 68.88%, while SS features are the least effective features at 58.74%.

### 5.3. Results for Hearing Impaired Case (NCM)

NCM is used as a performance metric for speech that has been processed by a vocoder in order to simulate speech perceived by a CI user. NCM values for babble and SSN at different SNRs are shown in Table 4. The NCM scores for SSN are better than those for babble as it is especially challenging for people with hearing impairment to listen in the presence of non-stationary noise. Table 4 shows that MS1 is best at −5 dB, MRCG is best at 0 dB and GF is best at 5 dB for babble. SS is worst at −5 dB, while GFCC is worst at 0 and 5 dB for babble. ES is best at −5 dB, CQCC is best at 0 dB and GF is best at 5 dB for SSN. GFCC is consistently worst for SSN. The MS1 feature mean NCM score for babble noise at −5 dB SNR is 0.5412. This is slightly better than the benchmark of 0.5343. However, its variation of 0.0720 is slightly worse than the benchmark variation of 0.0689. Compared with the benchmark, the feature vector is reduced in dimension from 246 to 48. The ES feature mean score for SSN at −5 dB SNR is 0.6972. This is slightly better than the benchmark of 0.6819. Its variation of 0.0302 is slightly better than the benchmark variation of 0.0329. Compared with the benchmark, the feature vector is reduced in dimension from 246 to 68.

Figure 10 shows the improvement in NCM relative to the unprocessed vocoded mixture (UN) for babble and SSN. Expressed as a percentage improvement in NCM at −5 dB SNR babble, this ranges from 37.41% for SS to 48.23% for MS1. The corresponding improvement for the benchmark is 45.39%. At −5 dB SNR SSN, this ranges from 44.57% for GFCC to 56.01% for ES. The corresponding improvement for the benchmark is 52.69%.

The score of 0.5366 for directly calculating MFCC is marginally better than using the wavelet approaches of SDWD (0.5321) and WMFCC (0.5267) for −5 dB babble. The corresponding score of 0.5272 for directly calculating PLP is marginally better than using the wavelet approach of WPLP (0.5252). The score of 0.6825 for directly calculating MFCC is marginally worse than using the wavelet approach of SDWD (0.6881) and the same as WMFCC (0.6825) for −5 dB SSN. The corresponding score of 0.6736 for directly calculating PLP is marginally worse than using the wavelet approach of WPLP (0.6888).

Figure 11 shows how NCM increases as SNR increases for both babble and SSN. The feature types plotted here were selected based on the worst- and best-performing mean NCM scores. When the NCM scores are further averaged for each feature type over all 3 SNRs and both noise types, the MS1 features are best at 0.7314. These are closely followed by MRCG features at 0.7304, while GFCC features are the least effective features at 0.6989. Further analysis of the distribution of performance scores across features, averaged across all test cases, shows the following broad behavior. Firstly, the average NCM across all the features is concentrated in the upper half of the observed range of NCM values, suggesting very small differences between the features. A similar trend is observed for STOI values, albeit with more of a spread across the observed range of STOI values. However, for HIT-FA, most of the features give performance in the middle of the observed range, with only one or two features at the upper end of the observed performance range for this metric.

### 5.4. Correlation of Metrics

Correlation coefficients between the various performance metrics were calculated across the various SNRs. Pearson correlation coefficient values can range from −1 to 1. Here, −1 represents a completely negative correlation, 0 represents no correlation and 1 represents a direct positive correlation. The results in Table 5 clearly demonstrate a strong positive correlation between STOI and NCM for both babble and SSN when evaluating the 18 different features in this study. The correlation is slightly higher for SSN than for babble. The correlation is less positive between HIF-FA and NCM for both babble and SSN when compared with STOI and NCM. However, in this case, the correlation is slightly more positive for babble than SSN. Overall, this indicates that the normal hearing metric of STOI is better than HIT-FA at predicting the intelligibility performance of acoustic features in hearing-impaired applications.

## 6. Conclusions

This paper has presented the results of a comparison of a range of existing features for speech enhancement in the context of cochlear implants. This includes features not traditionally used in this application. The speech enhancement method used herein utilizes the neural network architecture as described in [11]. It is worth noting that when shortlisting the features to be evaluated, PLP was chosen over RASTA-PLP for this study as it exhibited better intelligibility scores. The feature algorithms have been experimentally optimized in this study, and their parameters have been fine-tuned to maximize speech intelligibility.

As the main aim of this study was to focus on acoustic features, an MLP-based NN along with the IBM target were fixed elements of the system, based on their extensive study in prior literature; however, it is worth noting that many alternatives may be used.

The objective metrics chosen for this study were selected based on their ability to predict speech intelligibility. Speech intelligibility is of higher priority than speech quality in CIs. Further work could be carried out using metrics such as the Perceptual Evaluation of Speech Quality (PESQ) to examine the trade-off between intelligibility and quality. STOI score provides a reasonable indication of the ability of features to improve intelligibility for normal speech (speech that has not been processed by a vocoder), while NCM is used as a performance metric for vocoded speech. STOI scores were found to be higher for a tone vocoder than a noise vocoder. Although the results are relative, a noise vocoder was chosen for this study. There is quite a good correlation between STOI and NCM for babble and SSN. Therefore, it is reasonable to assume that the use of optimum features for normal hearing will also improve intelligibility for hearing-impaired listeners who wear devices such as cochlear implants in challenging listening environments. The correlation is not as good between HIT-FA and NCM.

The features were individually tested in this study in order to determine their individual contribution to improving speech intelligibility. However, it is obviously possible to combine groups of different features with the group lasso approach, as used in [76]. Because of the complementary nature of feature selection, features that result in high intelligibility individually may combine well with features that have lower intelligibility scores when used on their own. When benchmarking the results of this study against a feature vector used in [11], it was observed that certain features outscored or matched the group of features used in the benchmark in challenging, noisy conditions.

The optimum features depend on whether the noise is stationary or non-stationary and on the SNR. This information may be exploited in an adaptively configured speech enhancement system. On the basis of NCM scores in this study, MS1, MRCG and GF features would be selected for high, medium and low levels of non-stationary noise, respectively. Similarly, ES, CQCC and GF features would be selected for high, medium and low levels of stationary noise, respectively. The advantage of the proposed system is that only the feature set would change and not the architecture of the speech enhancement system. Further work would need to be conducted into the training implications when carrying out a detailed design of such a system. An obvious limitation of such a noise-adaptive architecture is that three different DNNs would have to be trained for three different levels of noise for each noise type, which would have implications for the resource requirements of the system, especially if the functionality is implemented on a resource-constrained edge device. Additionally, the system configuration depends on SNR, which can actually be difficult to accurately measure, especially in the presence of non-stationary noise.

The STOI and HIT-FA results of this study show that Gammatone Filterbank Power Spectra (GF) features provide the best overall intelligibility performance for normal-hearing listeners over a range of SNRs and noise types. Similarly, NCM results show that Mel Spectrogram (MS1) features provide the best overall intelligibility performance for hearing-impaired listeners over a range of SNRs and noise types.

It is also worth considering the computational load of the feature extraction block rather than that of the neural network itself. In this context, when compared with the benchmark group of features, the best-performing individual features (from those tested in this study) in challenging noisy conditions for normal and hearing-impaired applications (BS, ES, MRCG, GF and MS1) provided a reduction of the order of 30% in processing time for generating features and ideal masks. The exception to this was MRCG, which resulted in a substantial increase. The corresponding savings in training time was of the order of 25%. Again, the exception to this was MRCG, which resulted in a corresponding substantial increase. The above five features all fall into the category of frequency domain features.

It is also possible to perform wavelet decomposition and then extract features from appropriate coefficients/packets. However, when the noise is non-stationary, the results of this study indicate that there is no substantial benefit in terms of intelligibility of decomposing the speech signal and then extracting features such as MFCC/PLP over extracting the same features from the original speech signal. However, there may be marginal gains when the noise is stationary.

The speech samples used here are from male speakers and the English language; further work should use a wider variety of speakers and dialects. All testing was conducted here for matched-noise conditions only. Further tests could be done using unmatched noise conditions to test for good generalization with regard to speech segregation. Under these circumstances, the benchmark group of features may outperform the individual features used in this study. All testing performed here used a simulation of a cochlear implant and objective performance metrics for intelligibility. Further subjective testing could be conducted by presenting the vocoded speech to normal-hearing listeners.

The next stage in the development process would be to implement the prototype system on an experimental hardware research interface such as CCi-MOBILE described in [82]. This is a custom-made portable research platform that allows researchers to design and test speech-processing algorithms offline and in real time. Open-source software can be used to operate it, and it is compatible with cochlear implants from a leading manufacturer. It is designed on an FPGA platform and has a hardware processing pipeline for CI stimulation. Results for speech intelligibility in quiet and noisy environments have shown a consistent level of performance when compared with clinical processors of CI users [82].

Many of the features considered in this study have been shown to be robust for other applications, such as automatic speech recognition and classification-based speech separation. However, this study suggests that this does not guarantee that the same features necessarily improve speech intelligibility in noise for hearing-impaired listeners.

## Figures and Tables

**Figure 1 sensors-23-07553-f001:**
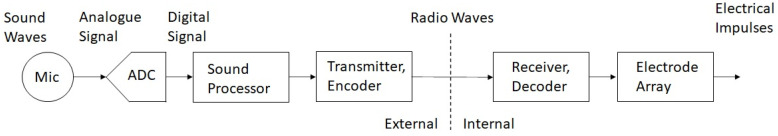
Block diagram of a cochlear implant.

**Figure 2 sensors-23-07553-f002:**
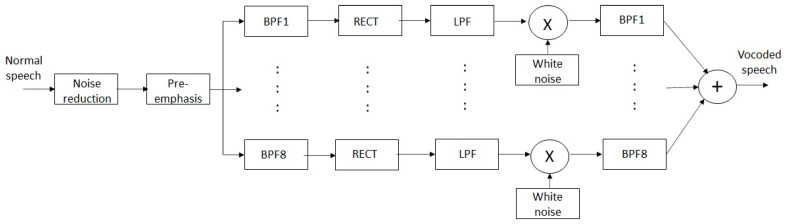
Block diagram of an eight-channel noise vocoder. BPF (band-pass filter), RECT (rectifier), LPF (low-pass filter).

**Figure 3 sensors-23-07553-f003:**
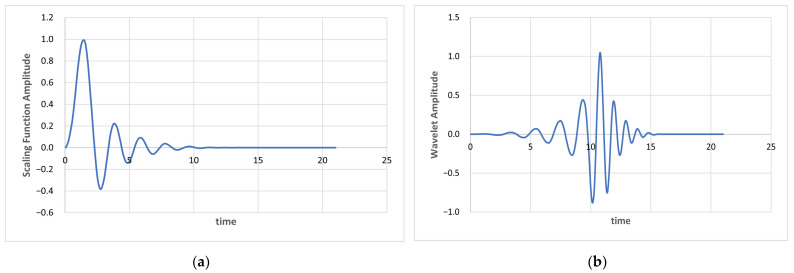
fk22 scaling function (**a**) and wavelet (**b**) used in this work.

**Figure 4 sensors-23-07553-f004:**
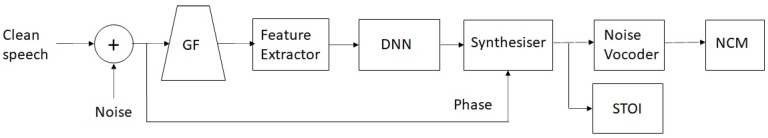
Block diagram of the test system. GF: Gammatone filterbank. DNN: Deep neural network.

**Figure 5 sensors-23-07553-f005:**
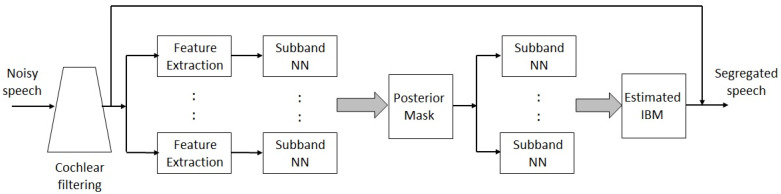
Block diagram of a speech segregation system.

**Figure 6 sensors-23-07553-f006:**
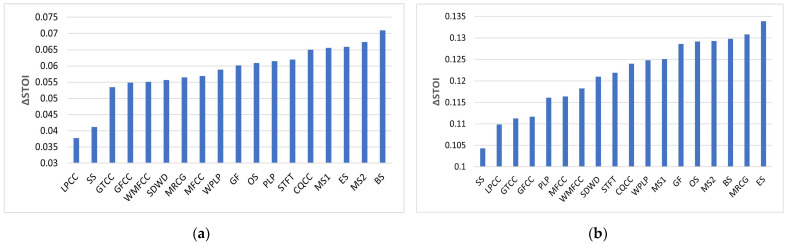
STOI improvement (ΔSTOI) relative to unprocessed mixture at −5 dB babble (**a**) and −5 dB SSN (**b**).

**Figure 7 sensors-23-07553-f007:**
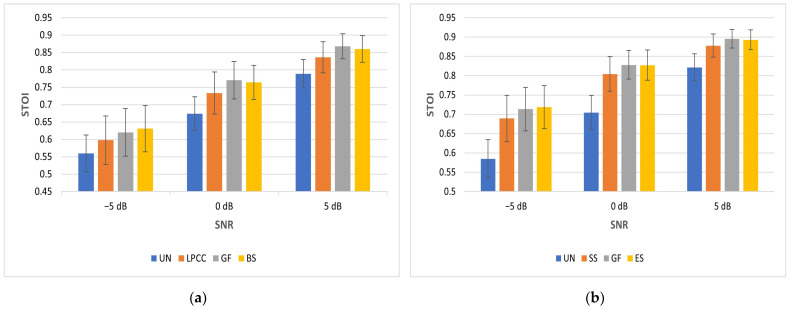
Mean STOI v SNR for babble (**a**) and SSN (**b**). Error bars show standard deviation.

**Figure 8 sensors-23-07553-f008:**
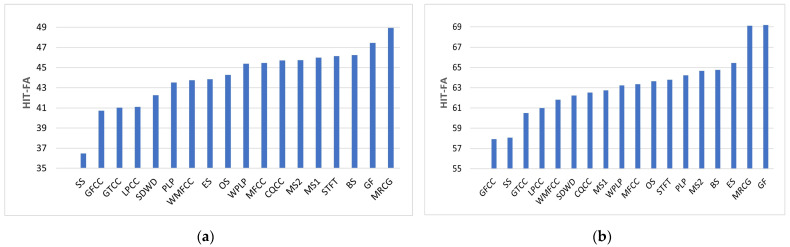
HIT-FA for −5 dB babble (**a**) and −5 dB SSN (**b**).

**Figure 9 sensors-23-07553-f009:**
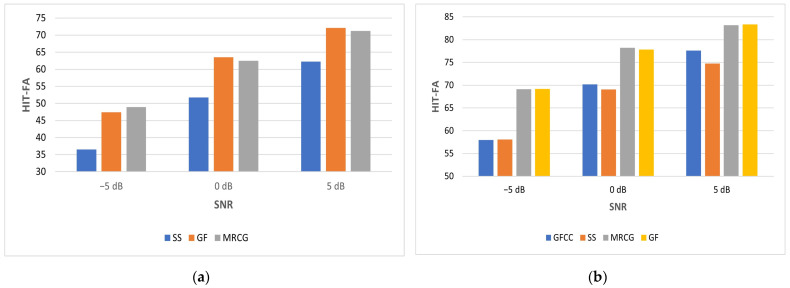
HIT-FA vs. SNR for babble (**a**) and SSN (**b**).

**Figure 10 sensors-23-07553-f010:**
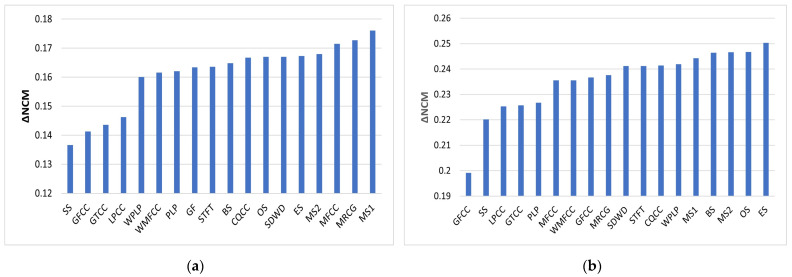
NCM improvement (ΔNCM) relative to unprocessed mixture for −5 dB babble (**a**) and −5 dB SSN (**b**).

**Figure 11 sensors-23-07553-f011:**
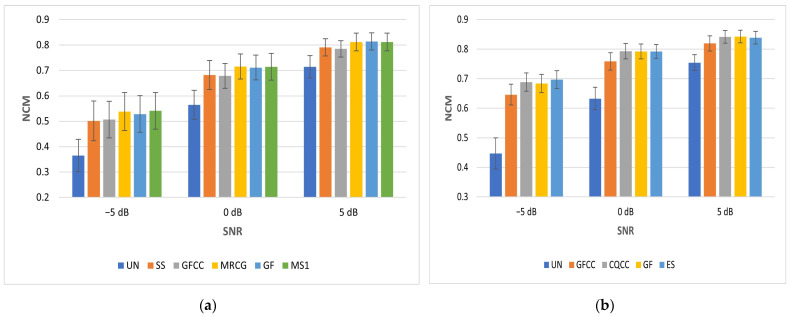
Mean NCM vs. SNR for babble (**a**) and SSN (**b**). Error bars show standard deviation.

**Table 1 sensors-23-07553-t001:** List of features in alphabetical order.

Feature	Abbreviation	Reference
Unprocessed mixtures	UN	-
Bark Spectrum	BS	Naing et al. [60]
Constant Q Cepstral Coefficients	CQCC	Todisco et al. [61]
Equivalent Rectangular Bandwidth Spectrum	ES	Naing et al. [60]
Gammatone Filterbank Power Spectra	GF	Wang et al. [11]
Greenwood Function Cepstral Coefficients	GFCC	Clemins et al. [62]
Gammatone Cepstral Coefficients	GTCC	Yin et al. [63]
Linear Prediction Cepstral Coefficients	LPCC	Chen et al. [64]
Mel Spectrogram	MS1	Turab et al. [65]
Mel Spectrum	MS2	Naing et al. [60]
Mel Frequency Cepstral Coefficients	MFCC	Alavi and Azimi [66]
Multiresolution Cochleagram	MRCG	Chen et al. [67]
Octave Spectrum	OS	ANSI [68]
Perceptual Linear Prediction Coefficients	PLP	Hermansky [69]
Sparse Discrete Wavelet Decomposition	SDWD	Dai et al. [70]
Spectral Shape	SS	Al-Shoshan [71]
Short Time Fourier Transform	STFT	-
Wavelet Mel Frequency Cepstral Coefficients	WMFCC	Palo and Kumar [72]
Wavelet Perceptual Linear Prediction Coefficients	WPLP	Gaafar et al. [73]

**Table 2 sensors-23-07553-t002:** Normal speech mean STOI for babble and SSN and standard deviation (in brackets). Best performing feature sets in bold and worst performing feature sets in italics.

Noise	Babble			SSN		
Feat/SNR	−5 dB	0 dB	5 dB	−5 dB	0 dB	5 dB
UN	0.5599 (0.0530)	0.6738 (0.0489)	0.7889 (0.0409)	0.5849 (0.0493)	0.7047 (0.0442)	0.8216 (0.0349)
BS	**0.6309** **(0.0666)**	0.7639 (0.0552)	0.8601 (0.0387)	0.7147 (0.0559)	0.8240 (0.0397)	0.8916 (0.0262)
CQCC	0.6249 (0.0636)	0.7613 (0.0547)	0.8622 (0.0379)	0.7089 (0.0557)	0.8197 (0.0404)	0.8917 (0.0260)
ES	0.6258 (0.0659)	0.7635 (0.0541)	0.8602 (0.0385)	**0.7188** **(0.0562)**	0.8273 (0.0394)	0.8928 (0.0258)
GF	0.6201 (0.0685)	**0.7704** **(0.0539)**	**0.8681** **(0.0362)**	0.7135 (0.0564)	**0.8281** **(0.0372)**	**0.8955** **(0.0239)**
GFCC	0.6148 (0.0639)	0.7616 (0.0521)	0.8626 (0.0363)	0.6966 (0.0598)	0.8191 (0.0411)	0.8910 (0.0257)
GTCC	0.6134 (0.0663)	0.7610 (0.0539)	0.8616 (0.0371)	0.6962 (0.0600)	0.8165 (0.0417)	0.8899 (0.0268)
LPCC	*0.5977* *(0.0698)*	*0.7333* *(0.0607)*	*0.8367* *(0.0447)*	0.6948 (0.0578)	0.8101 (0.0419)	0.8850 (0.0270)
MS1	0.6255 (0.0687)	0.7666 (0.0573)	0.8635 (0.0398)	0.7100 (0.0551)	0.8209 (0.0392)	0.8894 (0.0259)
MS2	0.6273 (0.0682)	0.7609 (0.0566)	0.8583 (0.0392)	0.7142 (0.0560)	0.8223 (0.0398)	0.8902 (0.0265)
MFCC	0.6168 (0.0669)	0.7611 (0.0544)	0.8610 (0.0378)	0.7013 (0.0571)	0.8170 (0.0405)	0.8902 (0.0267)
MRCG	0.6164 (0.0718)	0.7634 (0.0517)	0.8622 (0.0370)	0.7157 (0.0542)	0.8272 (0.0369)	0.8930 (0.0241)
OS	0.6208 (0.0667)	0.7660 (0.0540)	0.8615 (0.0392)	0.7141 (0.0544)	0.8201 (0.0388)	0.8894 (0.0261)
PLP	0.6214 (0.0639)	0.7571 (0.0541)	0.8572 (0.0393)	0.7010 (0.0549)	0.8192 (0.0400)	0.8895 (0.0266)
SDWD	0.6156 (0.0649)	0.7559 (0.0548)	0.8572 (0.0409)	0.7059 (0.0551)	0.8166 (0.0402)	0.8878 (0.0268)
SS	0.6011 (0.0654)	0.7436 (0.0557)	0.8455 (0.0402)	*0.6892* *(0.0600)*	*0.8045* *(0.0450)*	*0.8779* *(0.0301)*
STFT	0.6219 (0.0655)	0.7610 (0.0551)	0.8609 (0.0388)	0.7068 (0.0567)	0.8208 (0.0403)	0.8912 (0.0259)
WMFCC	0.6150 (0.0662)	0.7574 (0.0570)	0.8597 (0.0394)	0.7032 (0.0575)	0.8155 (0.0420)	0.8872 (0.0268)
WPLP	0.6188 (0.0669)	0.7588 (0.0564)	0.8584 (0.0401)	0.7097 (0.0568)	0.8194 (0.0402)	0.8890 (0.0263)

**Table 3 sensors-23-07553-t003:** Normal speech HIT-FA scores for babble and SSN. Epoch number (in brackets). Best performing feature sets in bold and worst performing feature sets in italics.

Noise	Babble			SSN		
Feat/SNR	−5 dB	0 dB	5 dB	−5 dB	0 dB	5 dB
BS	46.25 (17)	58.96 (13)	66.92 (13)	64.77 (12)	74.69 (19)	79.53 (19)
CQCC	45.71 (16)	59.55 (20)	69.68 (20)	62.53 (19)	74.02 (18)	80.68 (20)
ES	43.85 (17)	58.46 (17)	67.20 (16)	65.45 (16)	75.58 (20)	79.86 (20)
GF	47.45 (20)	**63.57 (14)**	**72.14 (14)**	**69.20 (14)**	77.80 (17)	**83.37 (16)**
GFCC	40.74 (20)	55.76 (18)	65.18 (18)	*57.94 (19)*	70.18 (20)	77.60 (20)
GTCC	41.04 (20)	57.85 (18)	67.42 (18)	60.50 (20)	73.20 (20)	79.46 (19)
LPCC	41.12 (16)	55.39 (16)	64.73 (15)	60.99 (16)	72.83 (16)	78.89 (20)
MS1	45.99 (20)	60.36 (20)	69.36 (20)	62.75 (13)	73.79 (19)	79.50 (19)
MS2	45.75 (17)	58.10 (17)	66.22 (19)	64.67 (12)	74.59 (19)	79.25 (19)
MFCC	45.46 (20)	60.29 (18)	69.61 (18)	63.35 (19)	74.85 (19)	81.07 (19)
MRCG	**48.95 (18)**	62.52 (16)	71.25 (20)	69.12 (10)	**78.21 (15)**	83.20 (20)
OS	44.28 (19)	59.08 (15)	67.34 (15)	63.64 (20)	73.89 (20)	79.12 (20)
PLP	43.52 (17)	57.55 (16)	67.42 (14)	64.24 (7)	74.49 (20)	80.53 (17)
SDWD	42.28 (17)	57.72 (14)	68.38 (19)	62.23 (17)	73.79 (17)	79.74 (17)
SS	*36.49 (19)*	*51.72 (15)*	*62.29 (19)*	58.08 (15)	*69.09 (20)*	*74.77 (20)*
STFT	46.14 (20)	59.96 (20)	69.44 (19)	63.79 (19)	74.79 (19)	80.84 (18)
WMFCC	43.76 (17)	59.33 (20)	69.42 (20)	61.83 (18)	73.30 (17)	79.80 (16)
WPLP	45.40 (16)	59.58 (20)	69.44 (15)	63.23 (18)	74.66 (18)	80.31 (18)

**Table 4 sensors-23-07553-t004:** Vocoded speech mean NCM for babble and SSN and standard deviation (in brackets). Best performing feature sets in bold and worst performing feature sets in italics.

Noise	Babble			SSN		
Feat/SNR	−5 dB	0 dB	5 dB	−5 dB	0 dB	5 dB
UN	0.3651 (0.0640)	0.5643 (0.0570)	0.7143 (0.0446)	0.4469 (0.0525)	0.6326 (0.0381)	0.7544 (0.0263)
BS	0.5299 (0.0732)	0.7055 (0.0506)	0.8042 (0.0350)	0.6933 (0.0288)	0.7903 (0.0255)	0.8380 (0.0229)
CQCC	0.5318 (0.0723)	0.7073 (0.0479)	0.8095 (0.0329)	0.6883 (0.0311)	**0.7929** **(0.0263)**	0.8415 (0.0217)
ES	0.5324 (0.0746)	0.7076 (0.0509)	0.8043 (0.0322)	**0.6972** **(0.0302)**	0.7922 (0.0231)	0.8387 (0.0217)
GF	0.5285 (0.0721)	0.7114 (0.0487)	**0.8139** **(0.0341)**	0.6836 (0.0311)	0.7919 (0.0254)	**0.8424** **(0.0216)**
GFCC	0.5064 (0.0717)	*0.6784* *(0.0488)*	*0.7848* *(0.0321)*	*0.6461* *(0.0352)*	*0.7583* *(0.0299)*	*0.8194* *(0.0258)*
GTCC	0.5087 (0.0765)	0.6957 (0.0502)	0.8037 (0.0337)	0.6726 (0.0335)	0.7801 (0.0249)	0.8339 (0.0218)
LPCC	0.5114 (0.0796)	0.6893 (0.0539)	0.7932 (0.0357)	0.6722 (0.0334)	0.7793 (0.0274)	0.8354 (0.0223)
MS1	**0.5412** **(0.0720)**	0.7146 (0.0524)	0.8118 (0.0349)	0.6912 (0.0304)	0.7924 (0.0249)	0.8374 (0.0196)
MS2	0.5331 (0.0722)	0.7072 (0.0510)	0.8049 (0.0346)	0.6935 (0.0287)	0.7902 (0.0254)	0.8376 (0.0228)
MFCC	0.5366 (0.0692)	0.7076 (0.0467)	0.8102 (0.0306)	0.6825 (0.0316)	0.7903 (0.0259)	0.8414 (0.0214)
MRCG	0.5378 (0.5378)	**0.7158** **(0.0494)**	0.8120 (0.0345)	0.6845 (0.0313)	0.7909 (0.0236)	0.8411 (0.0223)
OS	0.5321 (0.0749)	0.7078 (0.0535)	0.8035 (0.0346)	0.6936 (0.0320)	0.7902 (0.0244)	0.8338 (0.0226)
PLP	0.5272 (0.0784)	0.7025 (0.0530)	0.8041 (0.0362)	0.6736 (0.0320)	0.7910 (0.0250)	0.8409 (0.0197)
SDWD	0.5321 (0.0819)	0.7034 (0.0528)	0.8083 (0.0356)	0.6881 (0.0293)	0.7895 (0.0243)	0.8386 (0.0210)
SS	*0.5017* *(0.0784)*	0.6820 (0.0569)	0.7910 (0.0335)	0.6671 (0.0368)	0.7704 (0.0286)	0.8229 (0.0243)
STFT	0.5287 (0.0714)	0.7071 (0.0509)	0.8101 (0.0342)	0.6881 (0.0322)	0.7915 (0.0251)	0.8409 (0.0206)
WMFCC	0.5267 (0.0742)	0.7059 (0.0549)	0.8101 (0.0341)	0.6825 (0.0319)	0.7879 (0.0267)	0.8385 (0.0208)
WPLP	0.5252 (0.0759)	0.7014 (0.0532)	0.8084 (0.0356)	0.6888 (0.0301)	0.7890 (0.0253)	0.8397 (0.0221)

**Table 5 sensors-23-07553-t005:** Correlation of STOI vs. NCM and HIT-FA vs. NCM for babble and SSN.

Metrics	Noise	−5 dB SNR	0 dB SNR	5 dB SNR
STOI vs. NCM	Babble	0.9237	0.9519	0.9403
STOI vs. NCM	SSN	0.9855	0.9755	0.9641
HIT-FA vs. NCM	Babble	0.8386	0.8554	0.8872
HIT-FA vs. NCM	SSN	0.6164	0.7802	0.8235

## Data Availability

Speech utterances are from the IEEE-Harvard corpus and noise samples are from the Signal Processing Information Base.

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
