# Peer review of "Experimental Investigation of Acoustic Features to Optimize Intelligibility in Cochlear Implants"

_sensors, 2023, doi:10.3390/s23177553_

Round 1
Reviewer 1 Report
See attached file.

Reviewer 2 Report
This research has made a comparative investigation of various speech speech features experimented on speech enhancement method. And used perceptual quality estimation evaluation methods to demonstrate intelligibility , and showed experimental results, and made optimal feature suggestion.
This research has made a lot comprehensive experimental investigation, may be some discussions on underline reasons for the optimal feature suggestion could be beneficial for readers.
Reviewer 3 Report
The paper presents an analysis of a range of acoustic features for improving speech intelligibility in the context of cochlear implant applications. The paper presents a substantial amount of features for the aforementioned purpose and the results are interesting and useful for the community. Literature review is well presented. The experimental setup is somewhat limited. However, there is no statistical analysis on the results, and the use of a single male speaker is an important limitation of the study.
Minor comments:
1. The fact that wavelets did not work does not have to appear in the abstract.
2. Page 5, below figure. Applications of vocoders should be followed by appropriate references.
3. Spectral flux defintion is problematic. It should be like
SFt = (Σ |St(k) - St-1(k)|p)1/p
where t is the center of the current frame and t-1 is the center of the previous frame.
4. Check enumeration of Eq. 6.
5. Figure 3 does not add anything to the paper.
6. Having a single male speaker for this task is a major limitation of the paper. Why did the authors used that particular dataset? If any properties of the dataset are important for the task, it should be highlighted in the text.
7. Is dropout applied to all three layers of the DNN used for enhancement?
8. Results are nicely presented and discussed but I don't see any statistical analysis of the results. Are best methods significantly better than the rest? This is a severe limitation of the paper but I understand that this would probably extend the paper length too much. At least some analysis could be made for the best two features on each task.
None.
Reviewer 4 Report
The abstract could be improved with a more structured format, delineating the introduction, methodology, results, implications, and conclusion. It is missing details regarding the selection criteria for the chosen features, as well as the rationale behind employing specific metrics like STOI, HIT-FA, and NCM for intelligibility assessment. What is the feature size used in your study? Please specify. Why didn't the authors use the same epoch number to evaluate table 3? Adding a section that details the potential practical applications of these findings could enhance the paper's usefulness.
Round 2
Reviewer 1 Report
The paper notably improved after its revision. All of my comments were properly addressed by the Authors. Good job, well done!